# Pseudohypertriglyceridemia in a Patient with Pancreatitis Without Evidence for Glycerol Kinase Deficiency: A Rare Case Report and Review of the Literature

**DOI:** 10.3390/diseases13020029

**Published:** 2025-01-23

**Authors:** Jianping Zhu, Chunjuan Zhang, Rui Zhao

**Affiliations:** 1Department of Pharmacy, Sir Run Run Shaw Hospital, School of Medicine, Zhejiang University, Hangzhou 310016, China; zjping@zju.edu.cn; 2Department of Pharmacy, Haiyan People’s Hospital, Jiaxing 314399, China; 18842656942@163.com

**Keywords:** case report, diabetic ketoacidosis, glycerol, glycerol kinase deficiency, pseudohypertriglyceridemia, renal insufficiency, triglycerides

## Abstract

Background: Pseudohypertriglyceridemia (pseudo-HTG) is a condition in patients with glycerol kinase deficiency or other disorders of glycerol metabolism, as well as in individuals with alcoholism, severe liver disease, or metabolic disturbances, and those receiving heparin therapy. Exogenous glycerol intake can also trigger this condition. However, the causes of pseudo-HTG are poorly understood, and a clinical algorithm for its diagnosing remains to be developed. Case presentation: We present the case of a 46-year-old man admitted to hospital with hypertriglyceridemia-induced severe acute pancreatitis (HTG-SAP) and type 2 diabetes mellitus. Upon admission, his plasma triglyceride (TG) level was critically high at 43.78 mmol/L (3877 mg/dL). During hospitalization, he developed acute renal insufficiency and diabetic ketoacidosis (DKA). Despite conventional lipid-lowering treatments, including extracorporeal lipoprotein apheresis, his TG levels remained elevated. The unusually clear serum led to suspicion of pseudo-HTG. A glycerol-corrected TG assay confirmed normal TG values, thereby diagnosing pseudo-HTG. Conclusions: This report presents the first confirmed case of pseudo-HTG verified through definitive glycerol kinase (*GK*) gene testing in a patient without glycerol kinase deficiency. We also include a review of the relevant literature and propose a clinical algorithm. The case report highlights the importance of considering pseudo-HTG in hypertriglyceridemia patients who do not respond well to the standard TG-lowering treatment. Our proposed clinical algorithm for diagnosing pseudo-HTG is potentially invaluable in clinical practice, and helps to prevent unnecessary lipid-lowering treatments for patients with pseudo-HTG.

## 1. Introduction

Pseudohypertriglyceridemia (pseudo-HTG) is caused by high plasma glycerol levels that interfere with triglyceride (TG) measurements, and patients with this condition do not respond well to conventional lipid-lowering treatments. The true prevalence of pseudo-HTG remains uncertain, as identifying the actual rate is difficult due to the presence of asymptomatic cases. Discrepancies are observed between measured triglyceride levels and patients’ clinical manifestations, prompting an investigation into potential interfering factors [1]. The key to diagnosing pseudo-HTG is to test the TG levels using a glycerol-blanking assay, which emphasizes the need for clinicians to be aware of the condition [1]. However, the glycerol-blanking assay is not routinely implemented, leading to potential misdiagnosis.

The main causes of pseudo-HTG include glycerol kinase deficiency (GKD), metabolic disorders, severe liver diseases, and excessive glycerol intake [1,2], which can be categorized into endogenous and exogenous factors. Specifically, glycerol metabolism disorders, GKD, metabolic disorders, hyperthyroidism, and severe hepatic impairment constitute the endogenous factors [3,4]. Mutational analysis of GK genes can be considered for a definitive diagnosis of GKD. Although not routinely performed in clinical practice, genetic analysis may serve as an effective diagnostic tool, especially as testing becomes more widely available. In contrast, the intake of glycerol-containing medications, heparin therapy, or consumption of certain alcoholic beverages containing glycerol as an ingredient represents the exogenous factors [4,5]. Patients with pseudo-HTG often display clinical features such as increased TG levels in men, clear serum samples, decreased effectiveness of TG-lowering therapies, impaired glucose homeostasis or insulin resistance, and the absence of other metabolic syndrome features [3,4].

Herein, we report a case of pseudo-HTG secondary to severe acute pancreatitis (SAP), accompanied by severe renal insufficiency and DKA during hospitalization. Furthermore, we discuss the possible causes that led to pseudo-HTG in the present case and propose a clinical algorithm for diagnosing pseudo-HTG.

## 2. Detailed Case Description

The patient, a 46-year-old man, was admitted to Sir Run Run Shaw Hospital after recurrent episodes of abdominal pain lasting for two years, which had worsened for more than half a month. The patient denied consuming or abusing alcohol and was diagnosed as having hypertriglyceridemia-induced severe acute pancreatitis (HTG-SAP), along with abdominal bleeding, abdominal infection, and stage 3 acute kidney injury. The patient was diagnosed with acute pancreatitis two years ago, and following symptomatic treatment, his abdominal pain was significantly relieved. In addition, the patient had type 2 diabetes mellitus (T2DM) for one year, for which he took oral metformin, and he presented with good glycemic control. Furthermore, his thyroid function test results were normal.

At onset, the patient initially presented with severe upper abdominal pain that was persistent and radiated to the lower back without any obvious cause. Upon evaluation, it was found that the patient had an extremely high TG level of 43.78 mmol/L (3877 mg/dL). Furthermore, his serum lipase level was unusually high (821 U/L) (reference interval: <79 U/L). A CT scan showed diffuse enlargement of the pancreas with unclear boundaries, multiple low-density necrotic areas within the parenchyma, obscured peripancreatic fat spaces, along with a large amount of peripancreatic and abdominal fluid, indicating severe acute pancreatitis. The patient was administered a regimen at Lishui People’s Hospital, including pantoprazole to suppress acid, ustekinumab for anti-inflammatory effects, and octreotide to suppress pancreatic fluid secretion. However, on the second day after admission, the patient’s renal function deteriorated with an additional symptom of notable abdominal distension. Therefore, he was transferred to the intensive care unit for treatments including gastric decompression, plasmapheresis, continuous renal replacement therapy (CRRT), enema, and anti-infection therapy. Furthermore, a lipid-lowering treatment was administered, combining intravenous insulin at 0.3 units/kg/hour with low-molecular-weight heparin 5000 iu twice daily. Following five rounds of plasmapheresis, the patient’s TG level gradually decreased to 14.59 mmol/L (1292 mg/dL), and he was transferred to the general ward for further treatment. Despite this improvement, half a month later, the patient started showing the symptom of bilious vomiting, accompanied by distinct abdominal distension. A re-examination CT revealed that the patient’s acute pancreatitis had progressed, which prompted his transfer to our hospital.

By the time the patient was transferred to Sir Run Run Shaw Hospital, his TG level was 26.59 mmol/L (2355 mg/dL) and he was treated with extracorporeal lipoprotein apheresis. Since hospitalization, the patient had been fasting due to gastric decompression. On the 18th day after the onset of the disease, he was transferred to our hospital for treatment, at which point, his TG level increased to 30.31 mmol/L (2685 mg/dL), indicating that the previous treatment was not effective. The patient developed DKA, as evidenced by laboratory tests that showed an elevated blood glucose level of up to 20.0 mmol/L, beta-hydroxybutyrate level of 6.67 mmol/L (reference interval: 0.02–0.27 mmol/L), and a HCO_3_¯ level of 10.40 mmol/L (reference interval: 26–32 mmol/L). Subsequently, CRRT was administered, resulting in a decrease in the TG level to 18.96 mmol/L (1679 mg/dL), as observed on day 19. On day 20, his TG level again increased to 29.18 mmol/L (2584 mg/dL). Although ketoacidosis was corrected within 4 days of treatment, the TG level could be reduced only slightly to 22.85 mmol/L (2024 mg/dL). Interestingly, the patient’s plasma was clear, and his low-density lipoprotein cholesterol (LDL-C) level was notably lower (Table 1). Hence, the patient was suspected of having pseudo-HTG. To confirm the diagnosis of GKD, sequencing was performed on the specific region of the glycerate kinase (*GK*) gene, located at locus Xp21.3 of the X chromosome and encompassing 21 exons; however, the results were negative. Subsequently, the patient’s glycerol-blanked TG levels were tested, which were found to be 2.54 mmol/L (225 mg/dL), a value that was close to the normal range. Thus, the difference between these two values, which was 20.31 mmol/L (187 mg/dL), was considered to denote the patient’s serum glycerol level. Consequently, lipid-lowering therapy was discontinued, and the patient was transferred to the general ward. The trend of changes in his TG levels was evaluated in a reference laboratory (Figure 1). The patient recovered successfully, with no recurrence of pancreatitis during the 1-year follow-up period.

## 3. Discussion

To date, there have been limited studies reporting pseudo-HTG without evidence for GKD [4]. From 1995 to 2024, only a few studies on pseudo-HTG without evidence for glycerol kinase deficiency have been published and are summarized in Table 2 [4,5,6,7,8,9,10,11]. Among those ten patients previously described in the literature, eight (80%) of them were male and nine (90%) of them were over 40 years old. Common clinical features reported in these cases include male gender, HTG unresponsive to multiple lipid-lowering therapies, normal TG levels before onset, exogenous glycerol intake with or without impaired renal function, and clear serum samples. Additional clinical features include a normal body mass index (BMI) and inconsistent lipid profiles (normal cholesterol and high-density lipoprotein, etc.). Impaired glucose homeostasis has also been observed in individual cases. The age of onset in our cases is over 40 years old, with a medical history including pancreatitis, acute severe renal impairment, DKA, high BMI, and inconsistent lipid profiles. Moreover, the serum sample obtained from the patient appeared clear, and was consistent with the cases reported in the literature. However, none of the reported cases had been genetically tested for GKD. Given the varied clinical characteristics of GKD patients, genetic testing is vital for diagnosing GKD. Moreover, the underlying cause directly determines the subsequent treatment strategy. Our case is the first in which genetic testing for GKD was performed and GKD was definitively excluded.

Given the difficulties in diagnosing pseudo-HTG, we present a clinical algorithm in Figure 2, which can be used as a diagnostic aid in suspected cases. In HTG-SAP, therapeutic measures include insulin and/or heparin therapy, dietary changes, and other antihyperlipidemic medications [12]. Insulin increases the activity of lipoprotein lipase (LPL), which speeds up the breakdown of chylomicrons and lowers TG levels [13,14]. Heparin lowers the levels of TGs by releasing lipoprotein lipase that was originally stored in the endothelial cell [15]. In addition to these treatments, the patient was also treated with plasmapheresis as an emergency method of lowering TGs. However, in the present case, significant outcomes could not be achieved, and the patient’s TG level remained persistently high. Moreover, blood samples of patients with TG levels higher than 11.29 mmol/L (1000 mg/dL) may appear lipemic [16]. However, unlike the cloudy or celiac serum typically seen in true HTG cases, in the present case, the patient’s serum was clear. The glycerol-blanking procedure involves the enzymatic catalysis of free glycerol prior to TG hydrolysis [17]. During the glycerol-blanking procedure, the difference between the noncorrected and glycerol-corrected TG values indicates the concentration of free glycerol. Interestingly, the observed celiac serum at the beginning of the patient’s admission to the first hospital was different from the current presentation. The patient showed a true elevation in TG levels at the onset of pancreatitis.

To determine the etiology of pseudo-HTG, we investigated endogenous factors including GKD and the patient’s comorbidities. The most common cause of pseudo-HTG is GKD. GKD is a rare genetic disorder characterized by hyperglyceridemia and glycosuria, which are caused by the deletion or mutation of the *GK* gene [18]. GK catalyzes the phosphorylation of glycerol-to-glycerol-3-phosphate and plays a crucial role in glycerol metabolism and TG formation [19]. Most patients with GKD-induced pseudo-HTG show a normal body mass index (BMI) but increased TG levels (<11.29 mmol/L) [1]. However, the patient in the present study had a BMI of 33.12 kg/m^2^ and a consistently high TG level of >11.29 mmol/L, which did not align entirely with the clinical profile of GKD. Additionally, genetic testing ruled out mutations in the *GK* gene.

DKA may be an endogenous cause of pseudo-HTG, which cannot be excluded in the present case. There have been case reports indicating that diabetic ketoacidosis might result in HTG [20,21]. In cases of DKA with severe insulin deficiency, the elevated counterregulatory hormone triggers the activation of hormone-sensitive lipase in adipose tissue. This activation leads to the release of large amounts of glycerol and free fatty acids (FFAs) into the circulation when hormone-sensitive lipase lipolyzes endogenous TGs [22,23]. Free glycerol primarily comes from the lipolysis process in adipose tissue, indicating the mobilization of fat within the body. Under these circumstances, there is a significant increase in plasma free glycerol. Consequently, the excessive release of endogenous glycerol affects the accuracy of true TG measurement results.

In addition to DKA, severely impaired kidney function is also a potential endogenous factor contributing to pseudo-HTG. The patient also experienced renal insufficiency similar to a previously reported case [7], exhibiting acute renal failure with a decreased glomerular filtration rate of 11.29 mL/min/1.73 m^2^, which interfered with the normal excretion of glycerol. Research has demonstrated that the urinary excretion of glycerol commences when the blood glycerol concentration exceeds 1.6 mmol/L [24]. Furthermore, at blood glycerol concentrations ranging from 15 to 20 mmol/L, glycerol is predominantly excreted through urine [24]. However, due to the patient’s impaired renal function, glycerol could not be excreted completely through the urine.

The role of heparin cannot be ignored when considering exogenous factors leading to pseudohyperlipidemia. Heparin’s impact on TG metabolism is indeed a double-edged sword. Heparin can stimulate the release of lipoprotein lipase (LPL), enhancing its activity. This process accelerates the breakdown of chylomicrons and very-low-density lipoproteins (VLDL), thereby reducing the TG levels. However, according to a case study, a pregnant lady who used heparin for a long time acquired HTG-AP [25]. Research suggests that the serum TG level can rise again after a prolonged heparin infusion because heparin can reduce LPL on the surface of endothelial cells [26,27].

Exogenous glycerol comes from certain alcoholic beverages, pharmaceuticals, cosmetics (like lotions), cleaning supplies (such as soap and detergents), and parenteral nutrition. For this patient, there was neither alcohol intake nor parenteral nutrition. The only drug that the patient used that contained glycerol as an excipient was insulin. Glycerol is used as an excipient in certain insulin formulations to aid in stabilizing the medication and regulating its osmotic pressure. Specifically, this insulin solution generally contains 1.4 to 1.8 g of glycerol per 100 mL. During hospitalization in the first hospital, the patient consumed 23 bottles of insulin (400 IU, 10 mL/bottle, Wanbang Pharmaceutical Company, Xuzhou, China) in 15 days, containing 3.22–4.14 g of glycerol. This equates to a daily consumption of approximately 215–276 mg of glycerol. It has been established that a variation of ±1 mg/dL in the free glycerol concentration corresponds to a fluctuation of ±10 mg/dL in TG levels [1]. The intravenous administration of glycerol in the patient exceeded the oral intake documented in a previous case report [28]. The glycerol content in insulin among patients with renal insufficiency may not be overlooked. The patient’s exposure to glycerol through insulin formulations highlights the need to consider exogenous sources in pseudo-HTG cases.

This diagnostic algorithm is carefully crafted to determine the root cause of HTG using a methodical approach. The process begins with administering triglyceride-lowering treatments. If the patient responds well, it points to primary HTG. However, if these treatments do not work, further testing with glycerol-blanked triglyceride measurements is necessary. A normal result from this test suggests pseudo-HTG, which can be attributed to internal factors or external factors. Conversely, if the triglyceride levels stay high, secondary HTG, possibly due to severe illness or liver disease, is considered. This methodical approach facilitates precise diagnosis and allows for the efficient treatment of HTG.

Given its rarity and potential for underdiagnosis, the early detection of pseudo-HTG is crucial to avoid unnecessary tests and treatments. The patient’s TG level was determined using the glycerol-phosphoric acid oxidase peroxidase (GPO-POD) assay kit (Beckman Coulter Company, Suzhou, China). The TG method involves a series of coupled enzymatic reactions. In this method, TG in the sample is hydrolyzed by LPL to form glycerol and FFAs. Like the majority of TG tests conducted in clinical laboratories, the enzymatic analysis of TGs can be affected by glycerol interference. Without the implementation of a blanking procedure, an elevated level of free glycerol in a sample inevitably results in an increase in the measured TG level. The concentration of free glycerol is measured by the difference between the noncorrected and glycerol-corrected TG values. Thus, we used this procedure to determine the patient’s glycerol concentration, accounting for glycerol interference to compare the uncorrected and corrected findings for TG levels. In most cases, a glycerol-blanked TG assay is not clinically used as it involves unnecessary expenses; however, it is essential for patients suspected of having pseudo-HTG.

## 4. Conclusions

We present a rare case of pseudo-HTG without evidence of glycerol kinase deficiency. This case highlights the importance of considering pseudo-HTG among patients with clear serum samples who do not respond to conventional lipid-lowering therapy. Specifically, if patients also have metabolic disorders or renal insufficiency, as well as abnormally high TG levels and mismatched VLDL levels, the possibility of pseudo-HTG should be further evaluated. Diagnosing pseudo-HTG can be challenging because standard TG assays cannot determine the impact of free glycerol levels. In suspected cases of pseudo-HTG, we recommend using the glycerol-blanking procedure to determine the concentration of TGs and free glycerol. The timely identification of pseudo-HTG is critical for avoiding unwarranted medication and the potential overestimation of cardiovascular risks.

## Figures and Tables

**Figure 1 diseases-13-00029-f001:**
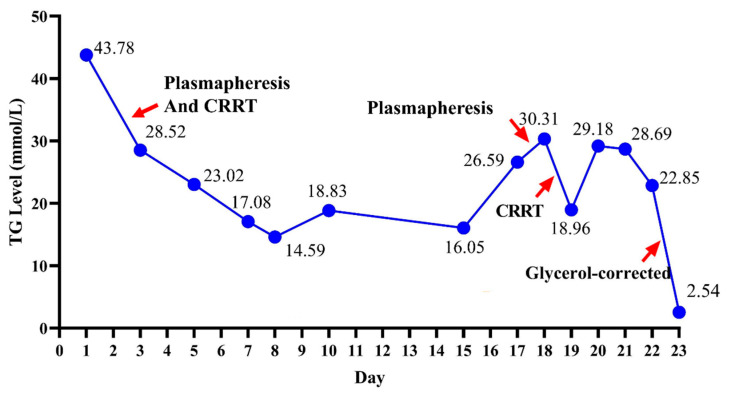
The changing trend in serum triglyceride levels. The arrows indicate the lipid-lowering treatment and the time points of the glycerol-corrected triglyceride assay. Abbreviation: CRRT, continuous renal replacement therapy.

**Figure 2 diseases-13-00029-f002:**
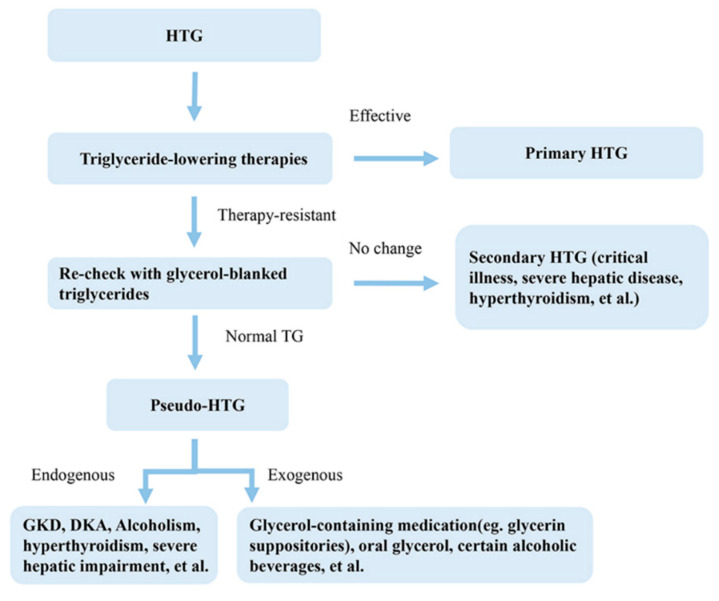
Clinical algorithm for pseudo-HTG diagnosis in suspected cases. Patients with pseudo-HTG do not respond well to standard lipid-lowering therapies. Evaluation of glycerol-blanked triglyceride levels helps to confirm the diagnosis of pseudo-HTG in such cases. Once the diagnosis is confirmed, both endogenous and exogenous causes must be identified. Abbreviations: HTG, hypertriglyceridemia; TG: triglyceride; GKD, glycerol kinase deficiency; DKA, diabetic ketoacidosis.

**Table 1 diseases-13-00029-t001:** Lipid measurements on day 18.

Parameter	Value	Reference Range
TG (GPO-POD)	22.85 mmol/L (928.86 mg/dL)	<1.70 mmol/L (69.11 mg/dL)
TG (glycerol-corrected)	2.54 mmol/L (103.25 mg/dL)	/
Glycerol, calculated	20.31 mmol/L (785.18 mg/dL)	/
VLDL-C (TG/5)	4.57 mmol/L (203.25 mg/dL)	/
LDL-C	0.97 mmol/L (37.51 mg/dL)	1.89–4.21 mmol/L (73.09–162.80 mg/dL)
HDL-C	0.68 mmol/L (26.29 mg/dL)	1.02–1.55 mmol/L (39.44–59.94 mg/dL)
Cholesterol	1.80 mmol/L (69.61 mg/dL)	3.00–5.70 mmol/L (171.6–326.04 mg/dL)

TG: triglyceride; VLDL-C: very-low-density lipoprotein cholesterol; LDL-C: low-density lipoprotein cholesterol; HDL-C: high-density lipoprotein cholesterol.

**Table 2 diseases-13-00029-t002:** Literature review of published pseudo-HTG cases that lack evidence of GKD.

Authors	Year	Age/Sex	BMI (kg/m^2^)	Medical History	Family History of HTG	Alcohol Use/Abuse	Clinical Characteristics	TG Level (mg/dL)	G-C TG Level (mg/dL)	Glycerol Level (mg/dL)	Serum Sample
Nauck M et al. [6]	1995	43/M	NA	a loss of hearing of the left ear; hypertension	NA	No	HTG with little response to therapy; healthy	2398	NA	158	clear
Speeckaert MM et al. [7]	2010	57/M	NA	ESRD; autosomal dominant polycystic kidney disease	No	4 drinks weekly	intake of glycerol-containing beer; a reduced glycerol clearance	996	NA	228	NA
Backes JM et al. [8]	2012	46/M	22	osteopenia secondary to hypercalciuria	No	2 ounces, 2–3 times weekly	HTG with little response to therapy; periodic TG levels reported as normal	405 to 552	88	NA	clear
Backes JM et al. [8]	2012	58/M	26	hypertension	No	No	HTG with little response to therapy; periodic TG levels reported as normal	381	NA	NA	clear
Arrobas-Velilla et al. [9]	2013	21/M	normal	none	cardiovascular disease	No	HTG with little response to fibrates	497–638	NA	370	clear
Backes JM et al. [10]	2013	60/M	27	none	NA	1–2 alcoholic drinks daily	HTG little response to therapy; healthy	488 to 532	49	NA	clear
Backes JM et al. [10]	2013	54/M	21	none	NA	No	HTG with little response to therapy; healthy	368 to 538	114	NA	clear
Backes JM et al. [11]	2015	52/M	NA	hypertension; a cerebral vascular accident	No	2 beers daily	HTG unresponsive to multiple treatments	552 to 695	62	NA	clear
Rajagopal R et al. [5]	2017	56/F	NA	T2DM; hypertension	No	3 to 4 cocktails daily	drank glycerol sometimes	1110	NA	NA	NA
Farooq A et al. [4]	2019	NA/F	NA	pancreatitis; T1DM; hypertension; ESRD; hypothyroidism; CHF	No	No	HTG with little response to therapy; ingestion of soap; impaired renal function	6620	151	6469	clear
Our case	2025	46/M	33.12	pancreatitis; T2DM	No	No	HTG unresponsive to multiple treatments; impaired renal function	928.86	103.25	785.18	clear

M, male; F, female; NA, information not available; ESRD, end-stage renal disease; CHF, congestive heart failure; T1DM, type 1 diabetes mellitus; T2DM, type 2 diabetes mellitus; HTG, hypertriglyceridemia; TG, triacylglycerol; G-C TG level, glycerol-corrected TG level; GKD, glycerol kinase deficiency; GK, glycerol kinase.

## Data Availability

All data analyzed during this study are included within the article.

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
