# Peer review of "Pseudohypertriglyceridemia in a Patient with Pancreatitis Without Evidence for Glycerol Kinase Deficiency: A Rare Case Report and Review of the Literature"

_diseases, 2025, doi:10.3390/diseases13020029_

Round 1

Reviewer 1 Report

Comments and Suggestions for Authors

Authors present an interesting case report for a rare condition and propose a clinically relevant diagnostic algorithm. To further strengthen the manuscript, I recommend emphasizing the unique aspects of this case, such as the role of genetic testing in excluding glycerol kinase deficiency. The discussion could be improved by reducing repetition. You should focus on the key takeaways, particularly the implications for clinical practice. Additionally, expanding on the practical application of the proposed diagnostic algorithm would enhance its utility for clinicians managing similar cases.

Author Response

Comment 1: Authors present an interesting case report for a rare condition and propose a clinically relevant diagnostic algorithm. To further strengthen the manuscript, I recommend emphasizing the unique aspects of this case, such as the role of genetic testing in excluding glycerol kinase deficiency.

Response: Thank you for pointing this out! I agree that it is important to emphasize the unique aspects of this case, particularly the role of genetic testing in excluding glycerol kinase deficiency. In the revised manuscript, I have added a more detailed discussion on the role of genetic testing in excluding glycerol kinase deficiency. This change can be found on page 2, line 46, in the revised manuscript.

“Mutational analysis of GK genes can be considered for a definitive diagnosis of GKD. Although not routinely performed in clinical practice, genetic analysis may serve as an effective diagnostic tool, especially as testing becomes more widely available.”

Comment 2: The discussion could be improved by reducing repetition. You should focus on the key takeaways, particularly the implications for clinical practice.

Response: We sincerely appreciate the valuable comments. In response, we have revised the discussion to eliminate redundant information and ensure that the main points are clearly and concisely presented.

Comment 3: Additionally, expanding on the practical application of the proposed diagnostic algorithm would enhance its utility for clinicians managing similar cases.

Response: Thank you for your insightful comment. In the revised manuscript, I have included additional details to illustrate how the algorithm can be effectively used in practice. This change can be found on page 7, line 223, in the revised manuscript.

“This diagnostic algorithm is carefully crafted to determine the root cause of HTG using a methodical approach. The process begins with administering triglyceride-lowering treatments. If the patient responds well, it points to primary HTG. However, if these treatments don't work, further testing with glycerol-blanked triglyceride measurements is necessary. A normal result from this test suggests pseudo-HTG, which can be attributed to internal factors or external factors. Conversely, if triglyceride levels stay high, secondary HTG, possibly due to severe illness or liver disease, is considered. This methodical approach facilitates precise diagnosis and directs efficient treatment of HTG.”

Reviewer 2 Report

Comments and Suggestions for Authors

The authors reported a rare case of pseudohypertriglyceridemia in patient with pancreatitis, who had no evidence of glycerol kinase deficiency. 

This report is well-written and may have potential interest. 

Minor point

Table 2 must be included the clinical features of the present patient. 

Author Response

Comment 1: Table 2 must be included the clinical features of the present patient.

Response 1: We sincerely thank the reviewer for careful reading. I have updated table 2 to incorporate the clinical features of the current patient, as recommended.

Reviewer 3 Report

Comments and Suggestions for Authors

Thank you for the opportunity to review the paper entitled. “Pseudohypertriglyceridemia in a patient with pancreatitis with-2 out evidence for glycerol kinase deficiency: a rare case report 3 and review of the literature”. I have read the manuscript with care and interest, I think it is well organized; however, to make it more attractive to potential readers, below are some comments for the authors to consider: Keywords are good to be arranged alphabetically. I am not sure if the citation method is appropriate for this journal (to be checked by the authors). It would be good to include epidemiological data and a very brief historical overview of the disease presented in the Introduction. There are editorial errors in the manuscript. The citation of Table 2 in the text is missing. Abbreviations used for the first time in the main text should be expanded, e.g. line 49 In conclusion, it is worth reducing the number of repetitions used. Grammar and punctuation corrections are worth making to the manuscript. More than 50% of the references are older than 5 years. References are not adapted under the requirements of the journal; in addition, it is worth adding the DOI number. Congratulations to the authors for an interesting manuscript.  

Author Response

Comment 1: Keywords are good to be arranged alphabetically.

Response 1: We sincerely appreciate the valuable comments. I have revised the manuscript to arrange the keywords alphabetically as recommended.

Comment 2: I am not sure if the citation method is appropriate for this journal (to be checked by the authors).

Response 2: Thank you for pointing out the issues with the citation method. I have carefully reviewed the journal's guidelines and have adjusted the citation style to align with their requirements.

Comment 3: It would be good to include epidemiological data and a very brief historical overview of the disease presented in the Introduction.

Response 3: We appreciate the reviewers' suggestions. Pseudohypertriglyceridemia is rare and its incidence is unknown; we have added an explanation of this in the introduction. Additionally, we have included a historical overview of the disease, covering its earliest descriptions, significant research advances, and its implications in clinical practice. The changes have been made on page 1, line 36.

“The true prevalence of pseudo-HTG remains uncertain, as identifying the actual rate is difficult due to the presence of asymptomatic cases. Discrepancies were observed be-tween measured triglyceride levels and patients' clinical manifestations, prompting an investigation into potential interfering factors [1].”

Comment 4: There are editorial errors in the manuscript. The citation of Table 2 in the text is missing.

Response 4: I appreciate your attention to detail. I would like to clarify that the citation for Table 2 has been included in line 124 at the appropriate location. However, I understand that it may not have been clearly visible.

Comment 5: Abbreviations used for the first time in the main text should be expanded, e.g. line 49. In conclusion, it is worth reducing the number of repetitions used.

Response 5: Thank you for pointing out the issues with abbreviations. I have expanded the abbreviations used for the first time in the main text, including the one mentioned on line 49. Additionally, I have reduced the number of repetitions throughout the manuscript. I appreciate your suggestions, which have helped improve the clarity of the text.

Comment 6: Grammar and punctuation corrections are worth making to the manuscript. More than 50% of the references are older than 5 years.

Response 6: Thank you for your feedback regarding the references. We enlisted an experienced language editor to enhance the manuscript and ensure it is free from grammatical errors. We acknowledge that a significant portion of our references are older than five years. However, these references are foundational and continue to be highly relevant to our study. They provide essential background and context that are critical to our research. We have ensured that the most recent and relevant studies are also included to provide a comprehensive view of the current state of research in this area. We hope this clarifies the rationale behind our choice of references.

Comment 7: References are not adapted under the requirements of the journal; in addition, it is worth adding the DOI number.

 Response 7: Thank you for your insightful feedback on the references. I've gone through and adjusted the reference list to match the journal's formatting guidelines. I've also added DOI numbers wherever possible. I truly appreciate your attention to detail and your helpful suggestions.

Reviewer 4 Report

Comments and Suggestions for Authors

The manuscript represents a rare case presentation. 

Authors should alsa define the criteria for severe acute pancreatitis. 

Some more imagistic considerations of CT findings could be detailed, and at least one suggestive image could be added. 

Any correspondence between the TG levels and the severity of pancreatitis could be establised? (https://doi.org/10.31689/rmm.2018.25.4.219)

Author Response

Comment 1: Authors should alsa define the criteria for severe acute pancreatitis.

Response 1: Thank you for your feedback. The diagnostic criteria for acute pancreatitis include the following three items: (1) Persistent epigastric pain; (2) Serum amylase and/or lipase levels at least three times the upper limit of normal; (3) Imaging findings consistent with acute pancreatitis on abdominal imaging studies. The diagnosis of acute pancreatitis can be established if two of the above three criteria are met.

The diagnosis of severe acute pancreatitis (SAP) is based on the presence of acute pancreatitis, complicated by persistent organ failure lasting for more than 48 hours. The diagnosis of organ failure is based on the Modified Marshall Scoring System, with an organ score of ≥2 indicating the presence of organ dysfunction.

Comment 2: Some more imagistic considerations of CT findings could be detailed, and at least one suggestive image could be added.

Response 2: Thank you for your valuable suggestion. The CT scans were conducted at an external hospital, and the results are stored in their information system. Due to confidentiality policies, we are unable to obtain the images. However, we have included a detailed description of the CT findings in the manuscript.

“CT scan showed diffuse enlargement of the pancreas with unclear boundaries, multi-ple low-density necrotic areas within the parenchyma, obscured peripancreatic fat spaces, along with a large amount of peripancreatic and abdominal fluid, indicating severe acute pancreatitis.”

Comment 3:Any correspondence between the TG levels and the severity of pancreatitis could be establised? (https://doi.org/10.31689/rmm.2018.25.4.219)

Response 3: Thank you for providing the literature, which allowed me to learn about the correspondence between TG levels and the severity of pancreatitis. The study found that hypercholesterolemia, particularly with cholesterol levels above 240 mg/dl, is a significant trigger for acute pancreatitis. Additionally, increasing cholesterol levels during the progression of acute pancreatitis can predict a poor outcome and evolution to organ failure, with high mortality rates. However, in this particular patient, the cholesterol level was not elevated, which is also one of the reasons why the patient was suspected of having pseudohypertriglyceridemia. We thank the reviewer for raising this issue, which gives us the opportunity to further refine our research in the future.